# Measuring the Motivation: A Scale for Positive Consequences in Pro-Environmental Behavior

**Insook Ahn [1,*] and Soo Hyun Kim [2]**

[1] Department of Family and Consumer Sciences, New Mexico State University, Las Cruces, NM 88003-8003, USA

[2] Retailing and Consumer Science, University of Arizona, Tucson, AZ 85721-0078, USA; sookim@arizona.edu

[*] Correspondence: iahn@nmsu.edu

**Abstract:** This two-part research study presents the development, validation, and assessment of a measurement scale designed to evaluate the positive consequences of pro-environmental behavior. Study 1 successfully constructed a scale with two factors: positive consequences for self (PCS) and positive consequences for the environment (PCE). Data collected from a diverse sample in the USA was analyzed using exploratory and confirmatory factor analyses, establishing the scale's reliability and construct validity. In Study 2, the nomological validity of the scale was examined, revealing significant relationships between values, positive consequences, anticipated positive emotions, and intentions to purchase pro-environmental products. Notably, biospheric and egoistic values were associated with different aspects of positive consequences, indicating the multifaceted nature of motivation for pro-environmental actions. Positive emotions mediated the impact of values and consequences on purchasing intentions. These findings provide valuable insights into the decision-making processes behind eco-friendly product purchases and contribute to understanding pro-environmental behavior. Future research can build on these findings to promote sustainable consumption in diverse contexts.

**Keywords:** awareness of positive consequences; positive emotion; pro-environmental behavior; value orientations





## 1. Introduction

The Value-Belief-Norm (VBN) model, initially developed by Stern and his colleagues, represents a foundational framework aimed at elucidating the determinants of individuals' environmentally significant behaviors and the motivations underpinning pro-environmental actions [1]. According to the VBN theory, pro-environmental behaviors are activated when individuals' personal norms or moral obligations are activated, typically in response to recognizing adverse consequences for objects they value and the associated attribution of responsibility. This theoretical paradigm has been broadly adopted across various academic domains, including but not limited to tourism [2], the hospitality industry [3,4], transportation [5–7], energy consumption [8], agriculture [9], among others.

Despite the substantial contribution of VBN-based research, particular areas for improvement have emerged. The construct of awareness of consequences (AC) with the VBN model has faced scrutiny due to concerns about its reliability and dimensionality problem [10,11]. Additionally, inconsistencies have arisen in the results of studies exploring the relationships between individuals' value orientations and their awareness of consequences. These discrepancies have urged some researchers to question the AC construct's validity and utility [10,12,13]. In response to these challenges, Ryan and Spash [10] suggested refining the AC scale, emphasizing the differentiation of beliefs regarding the positive consequences of pro-environmental actions and the perception of environmental harm.

The present study aims to address limitations associated with the AC construct within the VBN theory and delve into why individuals experience positive emotions when aligning their behaviors with their values.

## 2. Conceptual Background

This section will review the definition of AC and comprehensively explore the identified shortcomings of the AC construct, its intricate relationships with other constructs, and the specific research objectives guiding our investigation.

### 2.1. Awareness of Consequences (AC)

The VBN model, as developed by Stern et al. [1], employs a set of nine items to measure beliefs regarding consequences. These items are further categorized into three subsets: those relating to oneself (e.g., the better world both for the self and future generation, and security of one's livelihood), those on others (e.g., the quality of life and public health of others with the societal context), and those concerning biospheric considerations (e.g., preservation of endangered species and the dynamics of earth climate). This categorization assumes that individuals organize their beliefs about the negative consequences of environmental actions based on their endorsed value orientations. For instance, individuals who prioritize their self-interest will be concerned about environmental conditions threatening their well-being. In contrast, those emphasizing other human welfare will focus on the impact on others' health and quality of life. Similarly, those committed to non-human entities will be concerned about threats to these entities [14].

The VBN model posits that people are motivated to engage in pro-environmental behaviors based on the degree to which these behaviors may result in adverse consequences for objects they value. This motivation is calculated by aggregating the considerations across various value orientations [1].

### 2.2. Study Review: Pro-Environmental Behavior Incorporating with AC Consrtruct

In this subsection, we provide a comprehensive review focused on pro-environmental behaviors incorporating the AC construct. The aim is to critically evaluate its reliability within the VBN theory. Our compilation of relevant studies involved a thorough search on Google Scholar and an exploration of academic journals from 1990 to the present, identifying 39 academic studies that included the AC variable. Subsequently, we subjected these studies to a rigorous evaluation for inclusion in our review.

Out of the initial 39 studies, we employed a rigorous selection process, excluding those grounded in the Normative Activation Model (NAM) and its extensions, except for one study by Gärling et al. [6] featuring three distinct AC variables. Furthermore, we excluded studies lacking reliability values for AC. Our final selection consisted of 24 papers utilizing the AC construct as the primary measure for assessing beliefs linked to pro-environmental behavior.

Table 1 provides an overview of these selected studies, offering insights into the adoption of the AC variable, the composition of AC scale items, reliability values ($\alpha$, CR, and $\theta$), and the direction of survey questions.

**Table 1.** Comprehensive overview of studies adopting the AC variable.

| Study | Reliability | | Direction | Study | Reliability | | Direction |
| --- | --- | --- | --- | --- | --- | --- | --- |
| | AC Scales | AC Sub-Scales | | | AC Scales | AC Sub-Scales | |
| Han, H. (2015) [3] | 4 items (all bio), CR = 0.85 | | 3 (−), 1 (+), 1 R | Steg et al. (2005) [8] | 6 items (5 environment, 1 others; 4 specific, 2 general), α = 0.75 | | 2 (+), 3 (−), 1 R |
| Han et al. (2017) [15] | 3 items (all environment), CR = 0.90 | | 3 (−) | Snelgar (2006) [11] | 13 items (4 ego, 5 others, 4 bio) | AC bio (α = 0.46), AC alt (α = 0.56), AC ego (α = 0.30) | 4 (+), 4 (−), 5 R |
| Zhang et al. (2014) [16] | 3 items (2 environment, 1 our life), α = 0.64 | | 3 (−) | Jakovcevic and Steg (2013) [17] | 5 items (2 environment, 3 others), α = 0.66 | | 1 (+), 4 (−) |
| Raymond et al. (2011) [18] | 7 items (6 environment, 1 ego): Northern region: α = 0.54: SAMDB region: α = 0.68 | | 6 (+), 1 (−) | Lind et al. (2015) [19] | 6 items (items not available), α = 0.66 | | not available |
| Jansson et al. (2011) [20] | 5 items (4 environment, 1 other), α = 0.73 | | 2 (+), 3 (−) | Joireman et al. (2001) [21] | 13 items (4 ego, 5 others, 4 bio) | AC bio (α = 0.65), AC alt (α = 0.76), AC ego (α = 0.67) | 4 (+), 4 (−), 5 R |
| Kiatkawsin and Han (2017) [2] | 3 items (all environment), CR = 0.80 | | 3 (−) | Stern et al. (1999) [22] | 9 items (3 bio, 3 others, 3 ego), α = 0.88 | | 9 (−) |
| Wynveen et al. (2012) [23] | 6 items (2 environment, 2 others, 2 ego) | AC bio (α = 0.51), AC alt (α = 0.83), AC ego (α = 0.82) | 4 (+), 2 (−) | Gärling et al. (2003) [6] | 6 items (2 ego, 2 others, 2 bio) | AC bio (α = 0.54), AC alt (α = 0.42), AC ego (α = 0.45) | 6 (−) |
| Choi et al. (2015) [4] | 6 items (5 environment, 1 others), CR = 0.88 | | 2 (+), 3 (−), 1 (R) | Stern et al. (1993) [1] | Study 1: 9 items (3 bio, 3 others, 3 ego) | AC bio (θ = 0.56), AC alt (θ = 0.62), AC ego (θ = 0.66) | 1 (+), 4 (−), 4 R |
| Van Riper and Kyle (2014) [24] | 3 items (all environment), α = 0.85 | | 3 (−) | Nordlund and Garvill (2002) [25] | 12 items (bio, others, ego); items not reported; α = 0.82 | | not available |
| Hansla et al. (2008) [12] | 7 items (3 environment, 2 others, 2 ego) | AC bio (α = 0.56), AC alt (α = 0.56), AC ego (α = 0.64) | all (−) | Stern et al. (1995) [14] | 10 items (3 bio, 4 alt, 3 ego), θ = 0.85 | | 5 (+), 2 (−), 3 R |
| Zhang et al. (2020) [9] | 3 items (2 environment, 1 quality of life), α = 0.85 | | all (+) | Fornara et al. (2020) [26] | 2 items (1 alt, 1 ego), CR = 0.57 | | 1 (+), 1 (−) |
| De Groot and Steg (2007) [27] | 5 items (2 environment, 3 other), α = 0.81 | | 1 (+), 4 (−) | Hiratsuka et al. (2018) [28] | 5 items (2 env, 3 others), α = 0.65 | | 1 (+), 4 (−) |

alt = altruistic, ego = egoistic, AC = awareness of consequences, R = reverse direction.

Out of the total 24 studies, 50% (*n* = 12) reported low or moderate reliability (*α* or *θ* values below 0.70). Only six studies included three AC sub-scales (self, others, environment), and all reported low or moderate reliability. Eighteen papers employed an AC scale measuring a one-factor solution, yet four reported poor reliability (*α* < 0.70).

Regarding the use of reverse items used on AC construct, seven studies incorporated them, but only three reported acceptable reliability. Approximately 37% of the studies (*n* = 9) included all three value-related items (self, others, and environment), with three of these studies reporting good reliability (*α* or *θ* > 0.8) for the AC construct, where all items loaded onto a single AC factor. Additionally, eight studies employed questionnaires with the same directional items (one study employed positive direction, while seven used negative direction). Among these, three studies report low reliability, and four report acceptable reliability values.

The forthcoming sections detail the scale development process and the properties of the measure the extent to which individuals believe their environmental behaviors result in positive consequences for valued entities, such as themselves and the environment. The nomological validity of this measurement scale will be examined in the subsequent section.

## 3. Methodology

### 3.1. Preliminary Study: Item Generation

For the preliminary study, three distinct shopping scenarios were developed to elicit upward pre-factual thoughts regarding positive consequences and anticipated emotions stemming from potential actions. These scenarios revolved around apparel product purchasing situations, with each offering two t-shirt options: conventionally produced versus eco-friendly products. One scenario involved eco-friendly shirts prominently displayed at the front of the store. Participants were asked to assess the realism of the scenario and their intention to purchase an eco-friendly product. They were instructed to articulate pre-factual consequences and anticipated associated emotions from the purchase.

### 3.1.1. Procedure and Data Collection

Following the scenario description, participants were instructed to immerse themselves in the scenario and respond to questions concerning the realism of the scenario (assessed through three items on a 6-point Likert scale) and their intention to purchase an eco-friendly product (evaluated via a single item on a 6-point Likert scale). Additionally, participants were asked to delineate ten distinct pre-factual consequences resulting from their decision and to anticipate the associated emotions stemming from purchasing an eco-friendly product. Respondents were given ten empty spaces to articulate these consequences and positive emotions. This questionnaire was created by drawing inspiration from the antecedents and consequences of the future-oriented emotion model, as outlined by Baumgartner et al. [29] and Bagozzi et al. [30,31].

The present study utilized 88 participants out of the 90 recruited from an online consumer panel in the USA. Participants received monetary incentives for their involvement, and their ages ranged from 18 to 64, with varying percentages across age groups. Notably, 53.4% (*n* = 47) of the participants were male. The findings of this study corroborated the suitability and realism of the scenarios, with a high level of internal consistency observed (Cronbach's *α* = 0.90, *M* = 5.24, *SD* = 0.14). Participants reported that these scenarios closely mirrored situations encountered in their daily lives.

### 3.1.2. Analysis

An analysis of variance (ANOVA) indicated that the effect of different scenarios on the intention to purchase eco-friendly shirts was insignificant, $F_{(2, 85)} = 1.22$, *p* = 0.301. The respondents' intention to purchase eco-friendly products ranged from 1 to 6 (*M* = 4.41, *SD* = 1.29). Participants were instructed to provide various anticipated emotions (AEs) in response to the scenarios, which were subsequently identified and categorized as positive emotions associated with purchasing eco-friendly apparel products. Additionally, different

potential consequences of making such purchases were systematically coded into three primary categories: consequences for oneself (self), consequences for the environment (environmental), and consequences for others, including pro-environmental companies (others). Two coders meticulously reviewed the responses, and any discrepancies, which were rare (less than 2% of responses), were collaboratively resolved through discussion.

In total, the present study amassed a total of 278 responses through an open-ended questionnaire. To systematize the data, code frequencies were computed, and comprehensive summary reports were generated for each code. The primary categories from this coding process included consequences and positive emotions (see Table 2).

**Table 2.** Category, subcategory, and sample responses from preliminary study cording.

| Sub-Category (*n*) | Sample Responses |
|---|---|
| Environmental (137); Self (45); Other (6); No effect (35) | I can save the environment; Others might think highly of me; I could have a positive impact on society; It will not make much of environmental difference. |
| Feel better (27); proud (6); less or no guilty (6); moral (5); being a good person (4); happy (3) | I will feel better about myself; I feel proud about my purchase; I would probably feel less guilty; It is the moral thing to do; I am a good person; I would be happier with my purchase. |

Within the consequences category, participants' pre-factual consequences of their actions were classified into three primary subcategories: environmental (*n* = 137), self (*n* = 45), and other (*n* = 6). A subset of participants also believed that their actions would have no effect (*n* = 35). Sample responses within these subcategories provided valuable insights into participants' beliefs, such as "I can contribute to saving the environment", "Others might hold me in high regard", "I could have a positive impact on society", and "It will not significantly affect the environment".

Conversely, participants also articulated various positive emotions related to purchasing eco-friendly products within the second main category. The subcategories of positive emotions encompassed feeling better (*n* = 27), feeling proud (*n* = 6), experiencing reduced or negligible guilt (*n* = 6), feeling a sense of moral duty (*n* = 5), viewing oneself as a virtuous individual (*n* = 4), and feeling happiness (*n* = 3). Illustrative responses included statements such as "I will experience an improved sense of self", "I take pride in my eco-friendly purchase", "I would likely experience reduced guilt", "It aligns with my moral values", "I consider myself a morally upright person", and "It would bring me happiness".

Following selecting suitable items based on the generated pool and expert reviews, the research team refined the item list by carefully evaluating wording and redundancy. During this process, the focus was solely on consequences, and responses categorized as "no effect" were not further considered, aligning with the primary study objective of developing a scale to measure consequences for pro-environmental behavior. Ultimately, 14 items were chosen: seven items reflected positive consequences for oneself (PCS), five items pertained to positive consequences for the environment (PCE), and two items represented positive consequences for others (PCO).

The research team sought the input of five faculty members who possessed expertise in pro-environmental consumer behavior. These experts were asked to evaluate the relevance of each listed item and provide comments based on their understanding of the research team's intent. The review panel identified three critical aspects during their evaluation:

- Exclusion of PCO Items: The panel recommended removing two items from the PCO category, regarding them as irrelevant to the construct. These items were associated with companies that adhere to pro-environmental production and business practices, which did not align with the intended focus, human others.
- Elimination of Similar PCS Item: The panel suggested removing one item from the positive consequences for oneself (PCS) category, specifically the item related to the perception that others may notice the individual. This item was considered too similar to another existing item in the category.

- Lack of PCO Items: The panel noted the absence of items in the PCO category, highlighting the need to include items that investigate the phenomenon of positive consequences for others.

Considering the panel's feedback, the research team accepted the recommendation to remove the three items as proposed. Regarding the PCO items, in response to the expert panel's feedback, the research team carefully considered their inclusion in the measurement scale. The panel rightfully noted the absence of PCO items and emphasized the importance of investigating the phenomenon of positive consequences for others in the context of pro-environmental behavior. Upon thorough examination, the research team observed that only two PCO items were available for inclusion in the scale. Furthermore, these two items were found to be primarily associated with the positive consequences for companies that sell pro-environmental products, rather than directly addressing the impact on others, as conceptualized in the VBN theory.

With 11 items representing positive consequences (for self and the environment), the research team conducted an exploratory factor analysis (EFA) to ascertain the latent variables underlying this set of items. The subsequent section outlines the EFA process and presents the results obtained from Study 1. Additionally, the study discusses the evaluation of the scale's validity, encompassing convergent, discriminant, and nomological validity in Study 2.

### 3.2. Study 1: Refinement and Validation of a Scale for Accessing Positive Consequences in Pro-Environmental Behavior

Study 1 presents the development, refinement, and validation of a measurement scale designed to assess positive consequences associated with pro-environmental behavior. The study utilizes quantitative methods, including EFA and confirmatory factor analysis (CFA), to establish the reliability and validity of the scale.

### 3.2.1. Data Collection

Data for this study was collected through an online survey administered to a sample of 293 participants from the USA. The survey was distributed via the Amazon Mechanical Turk (MTurk) platform to ensure diversity in the participant pool. Demographic information was collected, including gender, age, education, income, and ethnic background. Of the survey participants, 46.4% of participants were male ($n = 137$), 52.2% were female ($n = 153$), and 1.4% were other ($n = 4$). In terms of age groups, 34.8% ($n = 102$) of the participants were between 18 and 35 years old; 23.9% ($n = 70$) were between 36 and 49 years old; and 26.6% ($n = 78$) were between 50 and 64; and 14.7% ($n = 43$) were over 65 years old. Most of the participants reported holding a bachelor's degree (37.9%, $n = 111$); approximately 29% ($n = 85$) reported they held high school education; 15.4% ($n = 45$) for associate degree; and 17.7% ($n = 52$) for graduate school degree. When asked about household income, a majority (35.2%) indicated that their income ranged from \$20,000 to \$49,999. In addition, about 32.1% of the participants' income was \$50,000–\$99,999, followed by an income of more than \$100,000 (17.4%), and less than \$20,000 (15.4%). In terms of ethnic background, about 68.9% of the respondents were Caucasian/White ($n = 202$); 14% were African American ($n = 41$); 7.5% were Asian ($n = 22$); 6.5% were Hispanic/Latino ($n = 19$); and 2.4% were of other ethnic background ($n = 7$).

### 3.2.2. Analysis

We used the scenario we demonstrated in the primary study for our preliminary study. After the scenario description, participants were instructed to put themselves in the scenario and respond to questions regarding their intention to purchase eco-friendly products (three items with a 7-point Likert scale), their beliefs about the realism of the scenario (three items with a 7-point Likert scale: 1 strongly disagree to 7 strongly agree). The reliability of the intention to purchase pro-environmental products was high ($\alpha = 0.902$, $M = 5.26$, $SD = 0.219$). Participants were then asked about the realism of a buying situation

scenario with two questions on a 7-point scale (1 strongly disagree to 7 strongly agree). The scale's reliability was high ($\alpha = 0.904$, $M = 5.23$, $SD = 0.134$), indicating that the scenario was highly reliable and realistic.

Scale purification and item refinement and validity were tested through exploratory factor analysis and confirmatory factor analysis.

EFA

EFA was conducted to refine the scale and assess its underlying structure using a principal component analysis with an orthogonal factor rotation method (Varimax) [32]. A Kaiser–Meyer–Olkin adequacy value of 0.919 and the high significance of Bartlett's test of sphericity ($\chi^2 = 2707.425$, $df = 55$, $p < 0.001$) confirmed the adequacy of the data for EFA. All indicators showed high communalities between 0.659 and 0.887. Initial Eigenvalues indicated that the first two factors explained 39.1% and 37.8% of the variance, respectively. The two-factor solutions explained 76.9% of the variance. All 11 items were loaded on two theoretical district constructs: PCS and PCE. Factor loadings for all items were above 0.77, and there was no cross-loading issue (Table 3). To assess the reliability of the solution, we examined the corrected item-to-total correlation and Cronbach's alpha values. The corrected item-to-total correlation was all above the threshold of 0.5 [33]. Cronbach's alpha values were 0.945 for PCE and 0.922 for PCS, respectively. The skewness and kurtosis for PCE and PCS were well within tolerable range for a normal univariate distribution [34].

**Table 3.** Results of the exploratory factor analysis for pre-factual consequences (*N* = 293).

| Item | PCE ($\alpha = 0.945$) | PCS ($\alpha = 0.922$) |
|---|---|---|
| PCE1 I am contributing to the healthy environment. | 0.921 | |
| PCE3 I will help sustain the earth longer. | 0.901 | |
| PCE2 I am helping curb environment problems. | 0.876 | |
| PCE5 I am doing my part in protecting the environment. | 0.868 | |
| PCE4 I am reducing my impact on the environment. | 0.856 | |
| PCS2 Others might think highly of me. | | 0.871 |
| PCS1 Others may notice me. | | 0.859 |
| PCS3 People will respect me. | | 0.854 |
| PCS5 It will set me apart from my peers. | | 0.805 |
| PCS6 I will have bragging rights. | | 0.801 |
| PCS4 Others will look at me as a champion of environmental causes. | | 0.769 |

Principle Component Analysis with Varimax rotation. Note: PCE = positive consequence for environment; PCS = positive consequence for self.

CFA

We conducted CFA using Maximum likelihood analysis with the Promax rotation method to assess the reliability and validity of the scale. All indicators showed highly significant factor loading in a range of 0.78 to 0.94 and thus met the threshold value of indicator reliability [35]. The CFA confirmed that the measurement structure of the positive consequences framework included an excellent fit to the data, with CMIN = 106.576, DF = 43, $\chi^2/df = 2.479$, SRMR = 0.047, CFI = 0.976, RMSEA = 0.071, and PClose = 0.021 [36] (see Figure A1). Composite reliability was calculated using factor loadings, which was significant at the 0.001 level. Internal consistency among multiple measurement items for each latent construct was evident in that all values for CR exceeded the minimum threshold of 0.60 [37]. Next, construct validity was tested. As shown in Table 4, AVE values range from 0.665 to 0.780, significantly above 50% [38]. square root of AVE values for each construct exceeded the correlations with another construct and AVE values are higher than MSV. In addition, we also tested the (HTMT). HTMT value was 0.461, significantly below the conservative threshold of 0.85 [39]. Accordingly, convergent and discriminant validity were supported.

**Table 4.** Inter construct correlation and reliability measures (N = 293).

|       | CR    | AVE   | MSV   | PCE       | PCS   |
|-------|-------|-------|-------|-----------|-------|
| PCE   | 0.946 | 0.78  | 0.206 | 0.883     |       |
| PCS   | 0.922 | 0.665 | 0.206 | 0.454 *** | 0.815 |

Note: CR = composite reliability; AVE = average variance extracted; MSV = maximum shared squared variance; PCE = positive consequence for environment; PCS = positive consequence for self. *** $p < 0.001$.

### 3.3. Study 2: Assessing Nomological Validity of a Scale for a Positive Consequence in Pro-Environmental Behavior

The motivation behind pro-environmental behavior is multifaceted, and one of the key motivators is the perception of positive consequences associated with such behavior. In Study 2, we delve into the nomological validity of a scale designed to measure these positive consequences. We focus on understanding how these consequences relate to individuals' value orientations, anticipated positive emotions, and intention to purchase pro-environmental products. This study is theoretically based on the VBN model [1] and the positive emotion theory [29,30].

#### 3.3.1. Procedure and Data Collection

To assess the nomological validity of our scale, we employed a survey-based method, similar to the approach used in our Study 1.

#### Measurement

We developed a structured questionnaire consisting of six sections for this study. Participants were presented with a scenario and asked to imagine themselves in that situation. They then completed questions about their intention to purchase pro-environmental products, their perceptions of positive consequences for themselves and the environment, their anticipated positive emotions regarding purchasing pro-environmental products, and their values. Additionally, two unrelated items were included in the survey to filter out unengaged respondents. The survey items to measure all constructs were adopted from previous literature for our study because of their proven reliability and validity, except for prefectural consequences construct measurements developed in Study 1.

Items for the pro-environmental product purchasing construct were assessed using self-reported measurements of buying experiences related to apparel products. Six items were adopted in previous literature [40,41]. Respondents rated each item using a five-point scale, ranging from 1 (never) to 5 (always) to specify how frequently they have purchased pro-environmental apparel within the last 12 months. These forms of pro-environmental product purchasing behavior were assessed using the frequency of consumers' self-reported buying activities instead of measuring actual engagement. The value orientation scales of De Groot and Steg [27] were adopted to measure three different types of respondents' value orientations: biospheric values (four items: respect for the earth, environmental protection, unity with nature, and pollution prevention), altruistic values (equality, world peace, social justice, and helpful toward others); and egoistic values (four items: social power, wealth, authority, and influence on others). All value items were put in a randomized order. Respondents rated the level of importance of each item as a guiding principle in their lives ranging from 1 (strongly unimportant) to 7 (strongly important). Positive consequences were captured in 13 items drawn from our 1st study: consequences for self (six items: e.g., Others might think highly of me) and consequences for environment (seven items: e.g., I am reducing my impact on the environment), and all items were put in a randomized order as well. Participants indicated the degree of agreement with the items on a seven-point scale, ranging from 1 (not at all) to 7 (Very much), the consequences of their purchase of environmentally friendly products. Anticipated positive emotions related to purchasing pro-environmental products (T-shirts) were measured on a 7-item Likert scale ranging from 1 (not at all) to 7 (very much). All anticipated positive emotion items (proud, relaxed, happy, excited, delighted, and glad) were adopted from Bagozzi et al. [42]. Finally, three items (e.g.,

I would consider an eco-friendly t-shirt when purchasing (next) clothing) for purchasing intention were assessed measure on a 7-item Likert scale ranging from 1 (strongly disagree) to 7 (strongly agree).

Data Collection

Data for the second study was collected through a web-based survey administered to 251 adult participants in the USA. After screening for the data quality, we included 244 responses in our analysis. The demographic characteristics of the participants included a gender distribution of 59.8% female, 39.8% male, and 0.4% others. Regarding household income, 39.8% indicated income ranging from $35,000 to $74,999, 33.5% had incomes less than $34,999, and 26.6% had incomes above $75,000. Regarding education, 52.1% reported holding bachelor's or graduate degrees, 26.6% indicated having a high school education, and 21.3% had a two-year degree. The ethnic background of participants included 80.3% White/Caucasian, 6.1% Hispanic/Latino, 6.1% African American, 4.9% Asian/Native American, and 2.5% other.

3.3.2. Analysis

EFA

EFA was conducted to evaluate the underlying structure of constructs in our study. The EFA employed the Maximum Likelihood extraction method with Promax rotation. The analysis showed that our data were suitable for EFA, as indicated by Kaiser–Meyer–Olkin adequacy value of 0.927 and a highly significant of Bartlett's test of sphericity ($\chi^2$ = 7918.622, *df* = 630, *p* < 0.001). All indicators exhibited high communalities between 0.428 and 0.945 except EV4 ("wealth" as a guiding principle in an individual's life). The eight-factor solutions, which explained 72.4% of the variance. All 36 items loaded on eight theoretically distinct constructs, including biospheric value (BV), altruistic value (AV), egoistic value (EV), positive consequences for self (PCS), and positive consequences for environment (PCE), positive emotions (PE), intention to purchase (IN), and purchase experiences (PUR). No cross-loading issues were observed.

CFA

Using a Maximum Likelihood extraction with Promax rotation, a CFA was conducted to confirm the measurement structure identified in the EFA. One item was removed to enhance the Average Variance Extracted (AVE) for egoistic value (EV4: wealth). Modification indices suggested covarying error terms between e16 and e17. Thus, we covaried them to achieve a better model fit. The CFA confirmed that the measurement structure of the values and positive consequences framework included an excellent fit to the data, with CMIN = 984.821, DF = 565, $\chi^2/df$ = 1.743, SRMR = 0.055, CFI = 0.946, RMSEA = 0.055, except PClose = 0.066 [36]. Composite reliability was calculated using factor loadings, which were all significant at the 0.001 level. Convergent validity was evident in that all values for composite reliability exceeded the minimum threshold of 0.60 [43]. Next, discriminant validity was tested. As shown in Table 5, AVE values range from 0.518 to 0.843. In addition, the square root of the AVE values for each construct exceeded the correlations with another construct except AV. Furthermore, the MSV for AV is bigger than that of AVE. Thus, we performed the HTMT ratio of correlations to ensure our model had no discriminant validity issue (see Table A1). All HTMT values were smaller than 0.850 for strict thresholds [39]. Accordingly, convergent and discriminant validity were wholly supported.

We delve into the comparison of the first and second-order models. The first-order model (8-factor model: BV, AV, EV, PCS, PCE, PE, IN, and PUR; see Figure 1) imposes fewer constraints than the second-order model (7-factor: BV, AV, EV, PCON (construct was combined with PCS and PCE to form the second-order model), PE, IN, and PUR; see Figure A2). As shown in prior studies, AC constructs were integrated into research models, with a majority of investigations opting for AC scales as a single construct, while others employed sub-scales (AC bio, AC alt, and AC ego), as illustrated in Table 2.

**Table 5.** Results of the measurement model (*N* = 244).

| | M | SD | CR | AVE | MSV | PE | PCS | PCE | PUR | BV | AV | EV | IN |
|---|---|---|---|---|---|---|---|---|---|---|---|---|---|
| PE | 4.74 | 0.342 | 0.948 | 0.723 | 0.419 | 0.85 | | | | | | | |
| PCS | 2.97 | 0.382 | 0.932 | 0.698 | 0.394 | 0.496 *** | 0.836 | | | | | | |
| PCE | 5.05 | 0.118 | 0.962 | 0.833 | 0.419 | 0.647 *** | 0.398 *** | 0.913 | | | | | |
| PUR | 2.27 | 0.34 | 0.883 | 0.656 | 0.236 | 0.426 *** | 0.335 *** | 0.350 *** | 0.81 | | | | |
| BV | 5.46 | 0.27 | 0.908 | 0.714 | 0.598 | 0.576 *** | 0.283 *** | 0.588 *** | 0.322 *** | 0.845 | | | |
| AV | 5.41 | 0.184 | 0.852 | 0.595 | 0.598 | 0.501 *** | 0.227 ** | 0.629 *** | 0.275 *** | 0.773 *** | 0.771 | | |
| EV | 3.18 | 0.375 | 0.763 | 0.518 | 0.394 | 0.355 * | 0.627 *** | 0.156 *** | 0.312 *** | 0.041 | 0.062 | 0.72 | |
| IN | 5.32 | 0.335 | 0.942 | 0.843 | 0.371 | 0.609 *** | 0.248 *** | 0.547 *** | 0.485 *** | 0.546 *** | 0.413 *** | 0.040 *** | 0.918 |

Note, PE = positive emotions; PCS = positive consequence for self; PCE = positive consequence for environment; PUR = purchasing experience; BV = biospheric value orientation; AV = altruistic value orientation; EV = egoistic value orientation; IN = intention. Square root of the AVE on the diagonal. * $p < 0.050$, ** $p < 0.010$, *** $p < 0.001$.

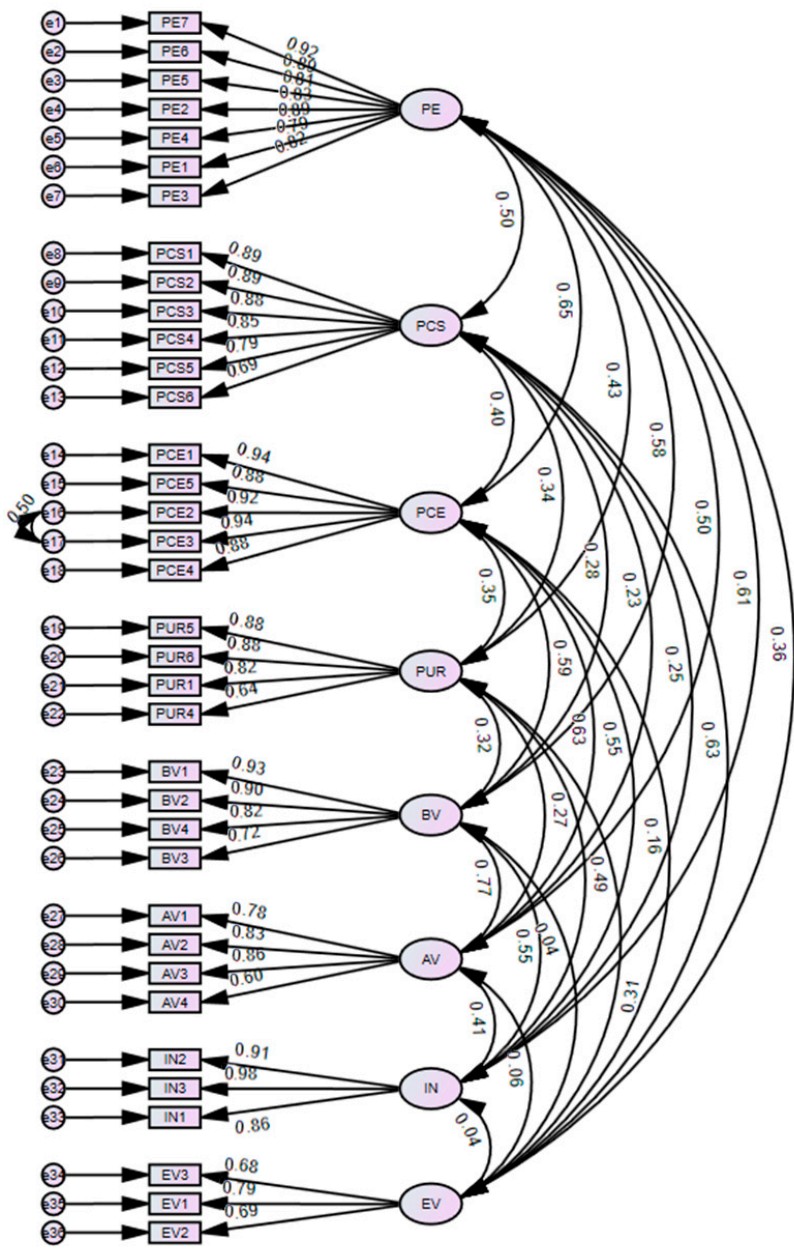

**Figure 1.** CFA for the first-order model. PE = positive emotions; PCS = positive consequence for self; PCE = positive consequence for environment; PUR = purchasing experience; BV = biospheric value orientation; AV = altruistic value orientation; EV = egoistic value orientation; IN = intention.

Recognizing the subtle nuances in model fit and the importance of an objective criterion for deciding between competing models, we conducted a $\chi^2$ difference test to compare different models (1st model and 2nd model) in terms of their fit ($\chi^2{}_{diff} = \chi^2{}_s - \chi^2{}_1$ and $df_{diff} = df_s - df_1$). This statistical test allowed us to determine whether the first-order model exhibits a significantly better or worse fit than the second-order model [44]. The $\chi^2$ difference was found to be significant ($\Delta\chi = 74.411$, $\Delta df = 5$, $p < 0.001$), indicating that the first-order model, with more freely estimated parameters, fits the data better than the second-model in which the relevant parameters are fixed. This decision aligns with our observation that the first-order model displayed superior fit indices for our data, while the second-order model, incorporating PCON, also encountered issues related to reliability and discriminant validity. Therefore, we chose to retain the first-order model for further analysis, despite the second-order model being a more parsimonious representation (see Table A2).

Validity Analysis

Before conducting the structural model test, efforts were made to enhance the measurement model's robustness. A common latent factor (CLF) was added to our CFA measurement model to capture any shared variance among all observed variables in the research model. A Chi-square different test was conducted to assess the effectiveness of CLF addition, comparing zero-constrained and unconstrained models. The results reveal that the constrained and unconstrained models were invariant, indicating that no specific response biases affected the model. Consequently, no distribution test for bias was deemed necessary, and the study proceeded to the structural equation model (SEM) phase.

In the SEM analysis, a control variable, prior purchasing experience, was included to account for any potential relationship between positive consequences, anticipated positive emotions, and the intention to purchase pro-environmental products. This step addressed the possibility that current self-expectations could activate pre-existing cognitive processes in memory, as suggested by previous research [45,46]. Then, to assess the nomological validity of our model, we examined the relationships among latent variables, including values and positive consequences, positive emotions, and purchasing (see Figure 2).

Our results showed that respondents endorsing biospheric values were more likely to perceive positive consequences for themselves ($\beta = 0.24$, $p = 0.019$) and for the environment ($\beta = 0.23$, $p = 0.015$) when considering the purchase of pro-environmental apparel. Surprisingly, AV orientation was not related to PCS but was statistically significantly related to PCE ($\beta = 0.41$, $p < 0.001$). Conversely, EV orientation was statistically significant and positively related to both PCS ($\beta = 0.61$, $p < 0.001$) and PCE ($\beta = 0.12$, $p = 0.047$). Importantly, EV exhibited more substantial predictive power for PCS than BV. AV emerged as the strongest predictor of positive environmental consequences, followed by BV and EV.

Controlling the purchase experience, we explored the relationship between value orientations, anticipated positive emotions, and intention to purchase (see Table 6). BV and EV were positively related to PE ($\beta = 0.32$, $p < 0.001$; $\beta = 0.18$, $p = 0.025$), while AV did not exhibit a significant relationship with PE. Both PCS ($\beta = 0.12$, $p < 0.001$) and PCE ($\beta = 0.37$, $p < 0.001$) were statistically significantly related to PE, which, in turn, was linked to IN ($\beta = 0.55$, $p < 0.001$). BV and EV orientations influenced IN but in different ways. BV was positively related to IN ($\beta = 0.26$, $p = 0.008$), whereas EV was negatively related to IN ($\beta = -0.16$, $p = 0.017$). PCE was directly related to IN ($\beta = 0.20$, $p < 0.001$), while PCS did not exhibit a significant direct relationship with IN.

**Table 6.** Standardized regression weights and squared multiple correlations for positive emotions and purchasing intention, and indirect effects of positive consequences on purchasing intention.

| | $R^2$ | $\beta$ | | $R^2$ | $\beta$ |
|---|---|---|---|---|---|
| DV: PCS | | | DV: PCE | | |
| BV | | 0.24 * | BV | | 0.23 * |
| EV | | 0.61 *** | EV | | 0.12 * |
| AV | 0.48 | −0.00 | AV | 0.46 | 0.41 *** |
| PUR [#] | 0.57 | 0.07 | PUR [#] | 0.54 | 0.13 * |
| DV: PE | | 0.32 *** | DV: IN | | 0.26 ** |
| BV | | 0.18 * | BV | | −0.20 * |
| EV | | −0.04 | EV | | −0.16 |
| AV | | | AV | | |
| PCS | | 0.12 | PCS | | −0.02 |
| PCE | | 0.37 *** | PCE | | 0.20 ** |
| PUR [#] | | 0.11 * | PE | | 0.37 *** |
| | | | PUR [#] | | 0.29 *** |

Note: * $p < 0.05$, ** $p < 0.01$, *** $p < 0.001$, [#] control variable. DV = dependent variable; BV = biospheric value orientation; AV = altruistic value orientation; EV = egoistic value orientation; PCS = positive consequences for self; PCE = positive consequences for environment; PUR = purchase experiences; IN = intention.

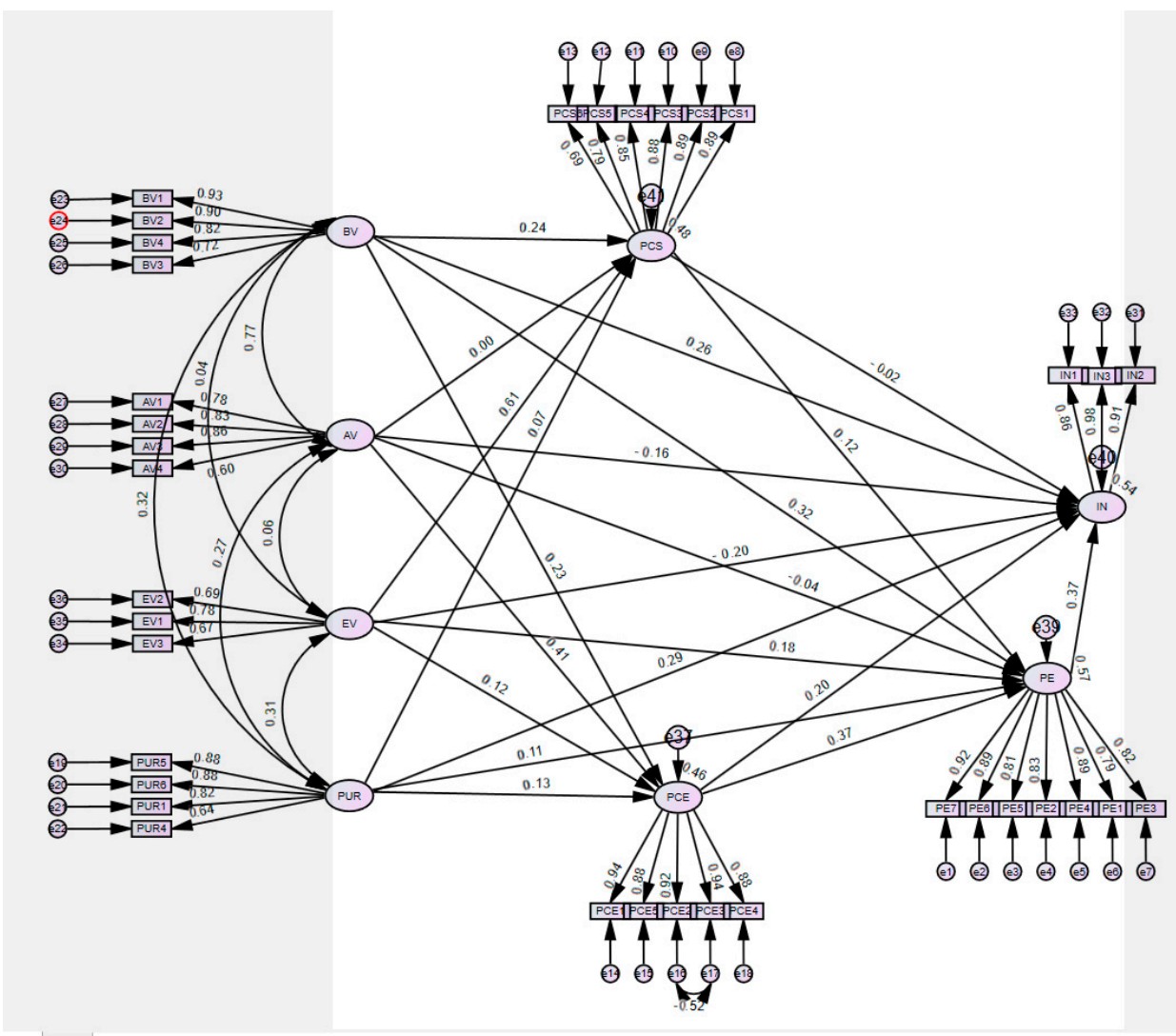

**Figure 2.** SEM analysis (CMIN/DF = 1.762, CFI = 0.944, SRMR = 0.058, RMSEA = 0.056, PClose = 0.043).

Indirect Effects of Variables

We employed the Hayes process macro to identify and analyze our research model's indirect effects and sequential mediations (Model 6). Standardized indirect effects were computed based on 2000 bootstrapped samples. We established 95% bias-corrected confidence intervals for these effects, as summarized in Table 7. The bootstrapped standardized indirect effects are followed. For the path from BV to PCE to PE, the indirect effect was 0.08 ($p < 0.05$) with a 95% confidence interval ranging from 0.015 to 0.207. For the path from BV to PE to IN, the indirect effect was 0.119 ($p < 0.01$), 95% confidence interval ranging from 0.036 to 0.241. In the case of the AV to PCE to PE path, the indirect effect was 0.150 ($p < 0.01$), with a 95% confidence interval ranging from 0.075 to 0.287. Similarly, for the AV to PCE to IN path, the indirect effect was 0.081 ($p < 0.05$), with a 95% confidence interval ranging from 0.011 to 0.201. For the path from EV to PE to IN, the indirect effect was 0.067 ($p = 0.01$), with a 95% confidence interval ranging from 0.022 to 0.182. Finally, for the PCE to PE to IN path, the indirect effect was 0.137 ($p = 0.002$), and the 95% confidence interval ranged from 0.041 to 0.232. These results indicate that all the indirect effects along these paths were statistically significant. Furthermore, when considering serial mediation effects, all three indirect pathways involving the influence of BV, AV, and EV on IN through PCE and PE were found to be significant ($\beta = 0.085$, $p < 0.05$, CI 0.005–0.095; $\beta = 0.150$, $p = 0.002$, CI 0.019–0.0137; $\beta = 0.044$, $p < 0.05$, CI 0.003–0.059, respectively).

**Table 7.** Indirect effects.

| Indirect Path | Lower | Upper | *p*-Value | SE |
|---|---|---|---|---|
| BV → PCE → PE | 0.015 | 0.207 | 0.032 | 0.085 * |
| BV → PE → IN | 0.036 | 0.241 | 0.006 | 0.119 ** |
| AV → PCE → PE | 0.075 | 0.287 | 0.003 | 0.150 ** |
| AV → PCE → IN | 0.011 | 0.201 | 0.049 | 0.081 * |
| EV → PE → IN | 0.022 | 0.182 | 0.010 | 0.067 ** |
| PCE → PE → IN | 0.041 | 0.232 | 0.002 | 0.137 ** |
| BV → PCE → PE → IN | 0.005 | 0.095 | 0.020 | 0.085 * |
| AV → PCE → PE → IN | 0.019 | 0.137 | 0.002 | 0.150 ** |
| EV → PCE → PE → IN | 0.003 | 0.059 | 0.034 | 0.044 * |

Note. * $p < 0.05$, ** $p < 0.01$, SE = standardized estimate; BV = biospheric value orientation; AV = altruistic value orientation; EV = egoistic value orientation; PE = positive emotions; PCE = positive consequence for environment; IN = intention.

These results provide valuable insights into our research model's complex relationships and mediation processes, highlighting the significance of positive consequences and anticipated positive emotions in influencing the intention to purchase pro-environmental products through various value orientations.

## 4. Findings

Through our preliminary study's item generation phase, we established the groundwork for developing measurement scales to assess positive consequences in the context of pro-environmental behavior. This process informed subsequent stages, including EFA and CFA, and the examination of relationships among value orientations, positive consequences, positive emotions, and purchasing intentions for pro-environmental products.

### 4.1. Deviation from VBN Theory

An intriguing departure from the theoretical underpinnings of the Value-Belief-Norm (VBN) theory emerged in the analysis of Personal Welfare and Concern for Others dimensions. Traditionally, VBN posits that individuals engaging in pro-environmental actions emphasize concerns related to well-being, progeny, and job security. Contrary to expectations, participants tended to prioritize self-enhancement in their pro-environmental acts, challenging anticipated correlations. Despite this deviation, participants' commitment to Environmental Considerations remained robust, underscoring the enduring importance

of ecological aspects. This suggests a complex recalibration of the perception of personal consequences, challenging the traditional conceptualization of AC. The observed shift toward self-enhancement and nuanced patterns in concern for others emphasize the evolving nature of individuals' environmental consciousness.

### 4.2. Measurement Scale Development

Our study successfully developed measurement scales for positive consequences (self-enhancement and for the environment) based on participant responses. These scales provide a robust tool for assessing the perceived benefits individuals associate with pro-environmental behavior.

### 4.3. Convergent and Discriminant Validity

The measurement scales demonstrated both convergent and discriminant validity, indicating their effectiveness in capturing distinct constructs related to positive consequences for self and the environment. Our findings suggest that the scales are reliable and valid for future research.

### 4.4. Relationships with Value Orientations

Exploring the relationships between different value orientations (biospheric, altruistic, and egoistic) and positive consequences revealed meaningful associations. Biospheric values were found to be positively associated with positive consequences for oneself and the environment. Altruistic values were related to positive environmental consequences, while egoistic values were associated with positive consequences for both oneself and the environment.

### 4.5. Positive Emotions and Intentions

Our research also delved into the role of positive emotions and their impact on purchasing intentions for pro-environmental products. Positive emotions were identified as mediators in the relationship between value orientations, positive consequences, and purchasing intentions. Fostering positive emotions related to pro-environmental behavior emerged as a potential enhancer of individuals' intentions to make eco-friendly purchases.

These comprehensive findings contribute to our understanding of the factors influencing pro-environmental behavior, particularly in the context of purchasing environmentally friendly products. The measurement scales' development and validation, exploration of relationships with value orientations, and insights into the role of positive emotions provide valuable contributions to the field.

## 5. Conclusions

Our study provides valuable insights into the motivational factors influencing pro-environmental behavior, specifically focusing on purchasing environmentally friendly products. The key points below summarize our findings and highlight their broader implications:

### 5.1. Summary of Findings

Our exploration of various facets of pro-environmental behavior, including the recalibration of consequences perception and the development of measurement scales for positive consequences, has significantly contributed to our understanding of these complex dynamics.

### 5.2. Contributions to the Field

Our study extends theoretical frameworks and offers practical applications for researchers and practitioners by shedding light on the complex interplay between personal and environmental considerations in pro-environmental decision-making.

### 5.3. Implications for Researchers and Practitioners

This study serves as a foundation for further investigations into the motivational factors driving pro-environmental behavior. Practitioners can leverage the validated measurement scales to understand individuals' perceptions and promote pro-environmental behaviors effectively.

### 5.4. Limitations and Future Directions

Acknowledging inherent limitations, we explore potential avenues for future research. The study contributes valuable insights, yet certain considerations and opportunities for further exploration should be recognized. Firstly, our focus on pro-environmental product purchasing behavior within a specific scenario may limit the generalizability of our findings. Future research endeavors could broaden the applicability of the developed measurement scales to encompass a broader spectrum of pro-environmental behaviors, such as energy conservation, recycling, or sustainable travel choices. Secondly, our concentration on positive consequences associated with pro-environmental behavior, while insightful, may benefit from an expansion of its scope. Future investigations might explore the integration of beliefs about adverse consequences (for oneself, the environment, and others) for not engaging in pro-environmental behavior. Thirdly, our exclusive focus on positive feelings in the context of purchasing environmentally sustainable products represents a deliberate narrowing of scope. Recognizing the suggestion to consider both positive and negative emotions, our future research agenda aims to address this limitation by incorporating both aspects into our model. This will enable a more comprehensive exploration of the emotional influences on consumer decisions regarding sustainable products. Lastly, our emphasis on positive consequences without considering awareness of consequences beliefs (both positive and negative) and personal norms, as in the original VBN model, is acknowledged as a limitation. Future research could involve a comparative analysis, assessing the proposed positive consequences model against the original VBN model. Such comparisons could offer insights into the relative importance of motivational factors in pro-environmental decision-making.

**Author Contributions:** Conceptualization, I.A. and S.H.K.; methodology, I.A. and S.H.K.; data analysis, S.H.K.; writing—original draft preparation, I.A.; writing—review and editing, I.A. and S.H.K.; funding acquisition, S.H.K. and I.A. All authors have read and agreed to the published version of the manuscript.

**Funding:** This research was funded by THE ASSOCIATION FOR CONSUMER RESEARCH, TRANS-FORAMTIVE CONSUMER RESEARCH.

**Institutional Review Board Statement:** All subjects gave their informed consent for inclusion before they participated in the study. The study was conducted in accordance with the Declaration of Helsinki, and the protocol was approved by the Ethics Committee of NMSU (18461).

**Data Availability Statement:** Details of the results from the individual test can be provided by sending a request to Ahn (iahn@nmsu.edu).

**Conflicts of Interest:** The authors declare no conflicts of interest.

## Appendix A

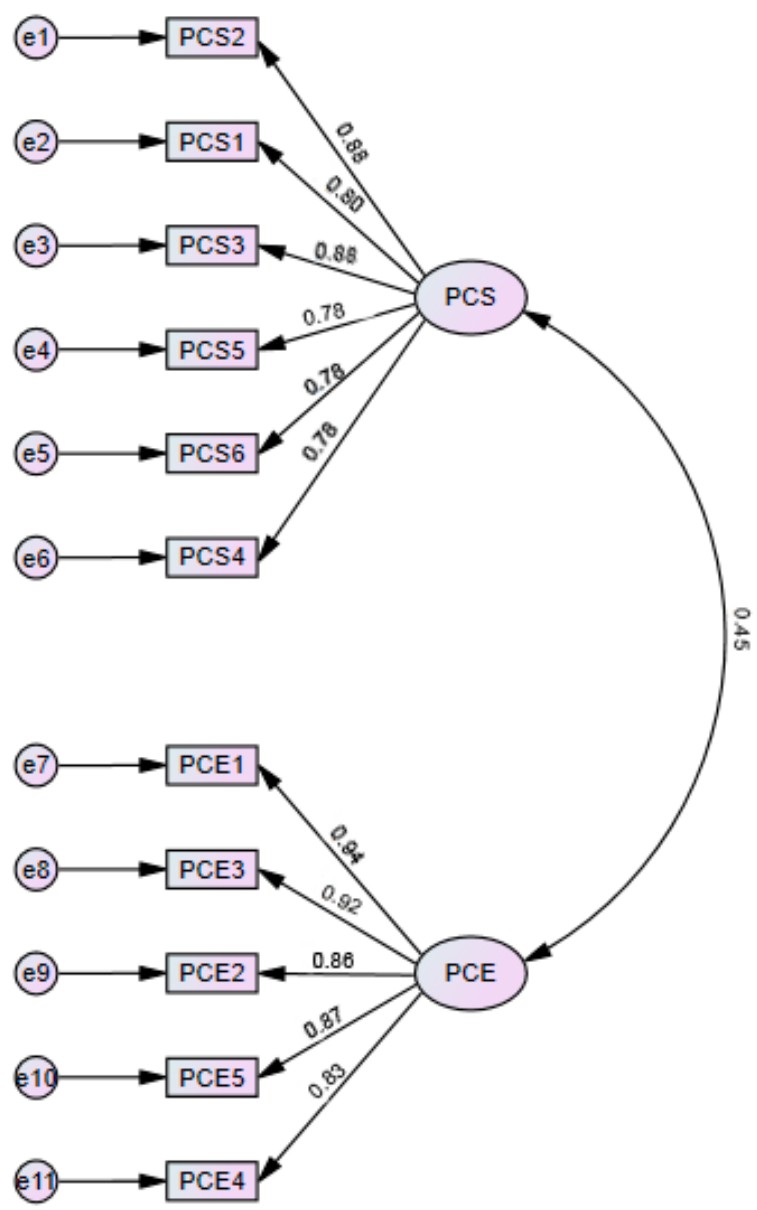

**Figure A1.** CFA, measurement structure of the positive consequences framework. PCS = positive consequence for self; PCE = positive consequence for environment. (CMIN = 106.576, DF = 43, $\chi^2/df$ = 2.479, SRMR = 0.047, CFI = 0.976, RMSEA = 0.071).

**Table A1.** HTMT analysis.

|  | **PCS** | **PCE** | **BV** | **AV** | **EV** |
|---|---|---|---|---|---|
| PCS |  |  |  |  |  |
| PCE | 0.404 |  |  |  |  |
| BV | 0.315 | 0.608 |  |  |  |
| AV | 0.22 | 0.64 | 0.768 |  |  |
| EV | 0.634 | 0.15 | 0.081 | 0.064 |  |

Thresholds are 0.850 for strict and 0.900 for liberal discriminant validity. PCS = positive consequence for self; PCE = positive consequence for environment; BV = biospheric value orientation; AV = altruistic value orientation; EV = egoistic value orientation.

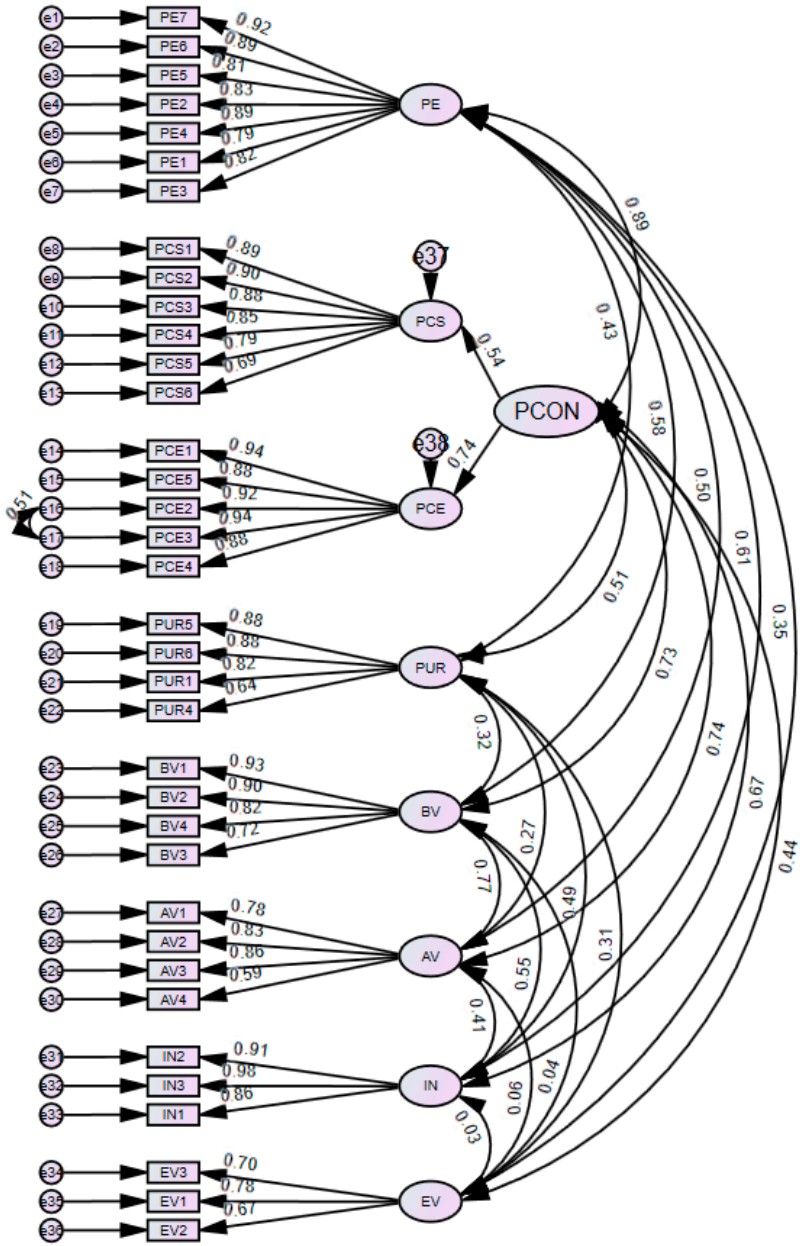

**Figure A2.** Seven-factor model fit.

**Table A2.** Model fit criteria for the eight-factor (1st order) model and the seven-factor (2nd order) model.

| Model | BV, AV, EV, PCS, PCE, PE, PUR, IN | BV, AV, EV, PCON, PE, PUR, IN |
|---|---|---|
| $\chi^2$ | 984.821 | 1059.232 |
| DF | 565 | 570 |
| $\chi^2/df$ | 1.743 | 1.858 |
| CFI | 0.946 | 0.937 |
| RMSEA | 0.055 | 0.059 |
| PClose | 0.066 | 0.003 |
| SRMR | 0.076 | 0.076 |
| $\Delta$ | $\Delta\chi = 74.411, \Delta df = 5, p < 0.001$ | |

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
