# Peer review of "Measuring the Motivation: A Scale for Positive Consequences in Pro-Environmental Behavior"

_sustainability, doi:10.3390/su16010250_

Round 1
Reviewer 1 Report
Comments and Suggestions for Authors
I have a few recommendations. First, congratulations on the interesting topic.
1. You started with an introduction which is not an introduction but a literature review mixed with some explanations regarding the methodology.
So, the introduction should be short and introduce the reader to the general topic and what you intend to research in your paper. Then, continue with a literature review where you present the variables in your study and how these are presented in other papers.
2. Table 1 should be presented at the beginning of the methodology as well as the explanation below it.
3. The section named Procedure should be Methodology. It is more common to use this title.
4. The Results and Discussion should be separate sections in the paper, not subsections for Procedure/Methodology
5. You do not have a clear vision or structure for what Methodology, Results and Discussions are. Because you have 2 studies, you should approach Methodology and explain the method for studies 1 and 2. Then it would be best if you continued with the results/findings for each study, where you present correlations and the other calculus.
6. After the results, you should present Discussions where you should add references for other studies that reached or did not the same conclusions as you did, for each of the two studies.
7. So, try to have one Methodology, one section for Results, and one for Discussions not several of them.
8. After Discussions, you should add a Conclusion section where you present limitations and future directions which you already have now and also the theoretical and practical implications of your paper (these and the novelty/originality of your research should be placed in conclusions before limitations).
Reviewer 2 Report
Comments and Suggestions for Authors
This two-part research study presents the development, validation, and assessment of a measurement scale designed to evaluate the positive consequences of pro-environmental behavior. Basically, VBN model is used and awareness of positive consequences is scrutinized. The paper highlights that positive emotions mediated the impact of values and consequences on purchasing intentions.
This is a well-written paper with proper methodology, lucid language, and extensive references. The figures are very well-done and the validity analysis is great. I only have brief suggestions and support the publication with very minor revision.
For the reader from other related disciplines with less familiarity with the terminology, I suggest adding a short appendix covering the terms such as AC – the awareness pf consequences, the basic definitions in VBN models such as difference in value and beliefs, etc.
The paragraph appearing the lines 132 to 135 is repeated in lines 140 to 143. One should be eliminated.
At line 189: An explanation would be nice to highlight how the expertise of the faculty in pro-environmental consumer behavior has been determined as this has an impact on the results.
At lines 206-209: It might be nice to do the evaluation without removing anything as an alternative and then comparing the results. This may also help to solve the expertise issue I highlighted above. I am not suggesting a new section but more like a footnote highlighting this issue.
Discussion section explains the findings lucidly and well-written.
In summary, this paper needs very minor revision and, in my opinion, is almost ready to be published in Sustainability.
Reviewer 3 Report
Comments and Suggestions for Authors
MANUSCRIPT TITLED:
Measuring the Motivation: A scale for Positive Consequences in Pro-Environmental Behavior
REVIEW:
Authors have carried out a full study focused on the development and validation of a measurement scale to evaluate the positive consequences of pro-environmental behavior. First of all, I would like to congratulate the authors for their work and encourage them to keep on working on this research line, which is currently a novel and very interesting topic which can be applied not only to buying decisions but also to other environmental behaviors.
I also would like to say that, obviously, this work is the result of a hard work carried out by the authors. However, I would like to provide a few comments and suggestions in order to help the authors to improve the quality of the manuscript prior to be published:
1. First of all, I have to say that, in general, the manuscript resulted hard to read for me due to the
I found it difficult to read due to the fact that there are often parts that are repeated throughout the text. For this reason, I would recommend that the authors revise the manuscript in its entirety in order to eliminate sentences or sections that are repeated.
Also, related to the general manuscript, I feel that it could be helpful for readers to include a summary table which summarizes the main characteristics of both studies 1 and 2: number of participants, objective, results, etc.
Likewise, it would be advisable to check the structure of the manuscript, avoiding making comments on certain parameters before they have been defined. Authors are requested to try to obtain a logical and linear structure.
As concepts appear for the first time in the text, the full name should be indicated together with the acronym, but this should only be done the first time it appears. The subsequent times it appears, only the acronym should be indicated.
Please modify Table 1 in order to avoid it taking up 3 full pages and to provide the information clearly.
Please unify the way of indicating decimal numbers, sometimes it appears as 0.9 and sometimes only .9, please unify the format preferably in the form 0.9.
Moreover, there are some typing mistakes. For example, in line 511, it is indicated that the effects are summarized in Table X, please, indicate the number of the corresponding table.
Between lines 206 and 208, it is indicated that it was decided not to include the items related to the OCPs, as they were initially derived from scenario-based conditions. However, it would be advisable to improve the justification.
There is an aspect related to the different tables included in this work, which does not make sense to me, since there are factors indicated in the text that are not included in the tables, such as PCON.
In section 2.3.3, when discussing the first and second order models, it would be advisable to indicate the equations of these models and to describe them in a more extensive manner.
Authors call section 3. Discussion. However, Discussion should include references, comparisons, etc. After reading this section, it is more correctly named 3. Conclusions.
I believe that since this study has focused only on positive feelings, it may condition the decisions made by the interviewees. If they had to write down not only positive but also negative emotions before making the decision about purchasing environmentally sustainable products, it is likely that the decisions would be different. For this reason, I would like to ask the authors for their reasons for not including both in the surveys.
Finally, in terms of format, I think that the format of the references section should be revised, as I believe that it does not coincide with that indicated by the journal.
Hope these suggestions are taken into consideration and help to improve the quality of this manuscript.
Thank you very much.
Sincerely.

Round 2
Reviewer 1 Report
Comments and Suggestions for Authors
I appreciate the improvement of your paper regarding the conceptual background and the argument regarding Table 1.
The structure looks better now but still confusing. After the method, you have findings and from your answer, I understand that you think are conclusions.
So, the Findings are similar to the results. Definitely not conclusions.
Please add a proper Conclusion section (not findings or discussions) after Findings. In conclusion, you should focus on the usefulness of your study for other researchers or practitioners and Limitations and Future research directions should be incorporated in the Conclusions, not separate.
Author Response
Thank you sincerely for your valuable

Reviewer 3 Report
Comments and Suggestions for Authors
Dear authors,
The manuscript has been highly improved. Thank you for taking into consideration my comments and suggestions, specially regarding the "Conceptual Background" and "Findings" sections. Table 1 is also very useful for readers.
Please, check the format of the manuscript, as for example in the Introduction section, the first and the second paragraphs show different line spacing.
Congratulations for the work you have carried out and good luck in the future.
Comments on the Quality of English Language
English use is fine.
Author Response
Response to reviewer is attached.
